



# Quality aspects of Fengyun3 D/E radio occultation bending angle products

Ying Li[1], Yan Liu[2][*], Wenwu Ding[1], Mi Liao [3], Xingliang Huo[1], Jinying Ye[1],

[1] State Key Laboratory of Precision Geodesy ,Innovation Academy for Precision Measurement Science and Technology, CAS, Wuhan 430077, China;

[2], China Meteorology Administration Earth System Modeling and Prediction Centre,  Beijing 100081, China;

[3] National Satellite Meteorological Centre, Chinese Meteorological Administration (NSMC/CMA), Beijing 100081, China.

*Correspondence to*: Yan. Liu (liuyan@cma.gov.cn)

**Abstract.** This study systematically evaluated the quality of ionosphere-corrected bending angles from Fengyun3 (FY3) D/E satellites (equipped with GPS and BDS receivers) using ERA5 data as references and MetOp products as comparisons. The quality of subsequent retrieved optimized bending angles, refractivity, and temperature were also analysed. Ionosphere-corrected bending angle were assessed via two approaches: outlier detection across 10–80 km and bias/noise quantification. Overall quality evaluation showed that FY3 ionosphere-corrected bending angles were consistent with MetOp below 40 km. Above 40 km, FY3 bending angles exhibited larger errors than MetOp. In outlier detection, MetOp had nearly 0% bad profiles, followed by FY3D (<10%), while FY3E (GPS/BDS) had about 20% bad profiles. FY3E-GPS bending angles are prone to have large outliers in the height range of 35-50 km. For bias and noise quantification, the daily mean biases and noise levels of FY3 satellites were higher than those of MetOp. Specifically, FY3E-GPS showed notable large daily mean biases of about -0.4 µrad and most of these biases are in rising RO events. FY3D and FY3E-BDS ranked second, with biases of approximately -0.1 µrad. MetOp had the smallest biases, at around –0.05 µrad. Regarding noises, FY3D, FY3E-GPS exhibited comparable noise levels, at roughly 2.5 µrad; FY3E-BDS had lower noises of 1.5 µrad. MetOp noises are smallest which are about 1.0 µrad. Due to the larger biases and noises at high altitudes, FY3's optimized bending angles were strongly corrected by background bending angles. Refractivity and temperature were also influenced by the strong correction of optimized bending angle. In summary, FY3 ionosphere-bending angles show high quality below 40 km. However, at high altitudes, further efforts are required to improve the quality issue for improving FY3 data's utility in numerical weather prediction and climate studies, especially for stratospheric applications.

## 1 Background

Global Navigation Satellite System (GNSS) radio occultation is a robust atmospheric remote sensing technique for sensing the Earth's atmosphere (Kursinski et al., 1997; Hajj et al., 2022). It provides vertical profiles, such as temperature, pressure, density and water vapour of the Earth's atmosphere. The basic theory of GNSS RO is to put RO receivers on the Low Earth Orbit (LEO) satellite to receive GNSS signals. As propagating through the atmosphere, GNSS signals will be bent due to





refractive gradient. Given the orbits of GNSS and LEO satellites, the formulated bending angle can be retrieved using geometric optic method (Melbourne, 1994). Since refractivity is related to bending angle, it can be calculated through the Abel transform. In dry air condition where the existence of can be neglected (Foelsche et al., 2008), temperature, density and pressure can be resolved using existing atmospheric relation equations in terms of the Smith-Weintraub equation, the ideal-gas law and the hydrostatic equation (Kursinski et al., 1997; Hajj et al., 2022). In moist air condition, additional background temperature/humidity profiles are required to be used for calculating atmospheric profiles (e.g., Healy and Eyre, 2000; Syndergaard et al., 2018; Li et al., 2019).

RO data has several distinctive advantages, such as high vertical resolution, high accuracy (in the upper troposphere and lower stratosphere), self-calibration, all weather condition and long-term consistency (Kirchengast, 2004; Steiner et al., 2011). Therefore, RO data has been widely used in numerical weather prediction, climate monitoring and space weather researches (Anthes, 2011). Positive impacts were received from these applications. For example, assimilation of RO data has significantly improved the performance of numerical weather prediction system (e.g., Cucurull et al., 2008; Healy and Eyre, 2000; Huang et al., 2010, Singh et al., 2021). With 11000 RO profiles being assimilated, RO is approaching being the most impactful dataset (Cardinali and Healy, 2014; Eyre et al., 2022). In the meanwhile, RO data have been proved to be useful in temperature trends detecting (e.g., Steiner et al., 2011, 2020), atmospheric boundary detection (e.g., Ao et al., 2012; Guo et al., 2011; Sokolovskiy et al., 2006), tropopause identification (e.g., Rieckh et al., 2014; Schmidt et al., 2010), sudden stratospheric warming events diagnostics (e.g., Li et al., 2021, 2023) and also tropical cyclones (e.g., Teng et al., 2023; Yang et al., 2023).

The first Low Earth Orbit (LEO) mission to employ the Radio Occultation (RO) technique was GPS/MET, launched in 1995 (Hocke, 1997; Ware et al., 1996). This mission successfully observed and retrieved highly accurate atmospheric profiles in the upper troposphere and lower stratosphere (Rocken et al., 1997; Steiner et al., 1999), thereby validating the concept of the GNSS RO technique for Earth's atmospheric sensing. Since 2000, a series of continuous RO missions have been launched, starting with the Challenging Mini-Satellite Payload (CHAMP) mission (Wickert et al., 2001), followed by the Gravity Recovery and Climate Experiment (GRACE) mission (Wickert et al., 2005). In 2006, the Constellation Observing System for Meteorology, Ionosphere, and Climate (COSMIC) mission—composed of six satellites—greatly advanced RO observations (Schreiner et al., 2007). Beginning in 2008, the European Meteorological Operational (MetOp) satellite program launched MetOp-A (Luntama et al., 2008), followed by MetOp-B in 2012 and MetOp-C in 2018. In 2013, China launched its first RO satellite: the Fengyun (FY) 3C (Liao et al., 2015, 2016a; 2016b, Sun et al., 2018). This satellite carried China's first domestically designed GNSS Occultation Sounder (GNOS) receiver. After FY3C, subsequent satellites in the FY3 series—FY3D through FY3G—have been gradually launched in recent years (Sun et al., 2023; Liu et al., 2023; He et al., 2023; Mo et al., 2024).

In addition to these satellites launched by institutional entities, a large number of commercial satellites have also been deployed. One of the earliest commercial RO satellites was launched by SPIRE in 2017. To date, SPIRE has launched approximately 60 RO satellites and now provides around 20,000 high-quality atmospheric profiles per day (Bowler, 2020;



Nguyen et al., 2022). PlanetiQ has launched five RO-related satellites and delivers over 6,600 high-quality neutral atmospheric profiles daily (Kursinski et al., 2024). In China, both the Tianmu and Yunyao satellite constellations have been put into operation: the companies behind these constellations have launched dozens of RO satellites, with more planned for future deployment (Xu et al., 2025). Collectively, these institutional and commercial satellites have created unique opportunities for weather and climate research.

In recent few years, the Radio Occultation modelling experiment (ROMEX), has been endorsed by the international radio occultation working group (Anthes, 2024). This experiment collects up to 35000 RO profiles per day of three months' period and tests the impacts of these data on numerical weather prediction system. The data are available to all users for testing in global and regional models, and they are also available for other scientific studies in weather and climate. Initial results found good impacts on NWP system (e.g., Anlauf et al.,, 2024; Li, et al., 2024; Lonitz et al, 2024) and on improving

forecasts of Atlantic hurricanes (Miller et al., 2024).

The purpose of our work is to further evaluate and improve data quality of Fengyun series RO satellites in stratospheric region. As a first step, this study works on analysing the quality aspects of Fengyun RO ionosphere-corrected bending angle in terms of data quality as well as biases and noises. There are current a lot of work have analysed the quality of Fengyun series RO satellites. Liao et al., 2016b has validated RO refractivity of FY3C satellite. Results showed that its refractivity

quality were similar to COSMIC and MetOp in the height range of 0-30 km. In addition, it also points out that FY3C products needed further improvement above 30 km. Sun et al., 2018 and Bai et al., 2018 further validated FY3C atmospheric and ionospheric data. Liu et al., 2023 evaluated FY3E GNOS II RO retrievals of BDS system. He et al., 2023 evaluated the quality of atmospheric radio occultation profiles from FY3E/GNOS-II BDS and GPS measurements. Currently, GNOS-II has been carried by FY3E, FY3F and FY3G. Liu et al., 2024 used an improved three-cornered hat method to evaluate the

FY3E moist retrieval data using model data and radiosonde measurements. Mo et al., 2024 evaluated performance of multi-source GNSS radio occultation from COSMIC-2, MetOp-B/C, FY3D/E, SPIRE and PlanetiQ over China. In addition to these published papers, ROMSAF has published a couple of reports in analysing FY3C and FY3E data quality (Bowler et al., 2019, Lewis et al., 2025).

Although, there are already many existing evaluation works, many key aspects have never been discussed yet, i.e., bending

angle quality. Bending angle, especially ionosphere-corrected bending angle, is one of the original RO measurements which is not influenced by background models in the process like statistical optimization and 1dVar (Li et al., 2013, 2019). Understanding the quality of ionosphere-corrected bending angle are useful for understanding the quality of subsequent retrieved RO products. However, the quality of bending angle has rarely been discussed before, especially above the middle stratospheric layers (>30 km). In order to further improve Fengyun data quality and get ready for the NOMEX experiment

for more widely numerical weather prediction applications, it is important to improve Fengyun atmospheric profiles above middle stratospheric layers. Therefore, this research works on examination the quality of Fengyun ionosphere-corrected bending angle and their influences on subsequent retrieved RO profiles in terms of optimized bending angle, refractivity and temperature. This study is a joint work with Chinese Meteorology Administration (CMA) aiming at improving the



performance of Fengyun data in its NWP system. It is expected that the outcome of this study and also our subsequent
research are useful for further enhancing NWP system of CMA and for improving the application of Fengyun data in
stratospheric climate applications.

## 2 Data and methodology

### 2.1 Data

In this study, we used Fengyun-3D (FY3D), Fengyun-3E GPS (FY3E-GPS), Fengyun-3E BDS (FY3E-BDS) and MetOp RO
ionosphere-corrected bending angle observations. The data period is August 2023. Fengyun data we used is one of the latest
version of Fengyun reprocessed dataset provided by CMA. It is also known that the Fengyun data has been reprocessed
again recently to improve its data quality below 30 km and positive impacts have achieved. However, this does not affect our
results in this study since we focus on bending angle quality above 30 km. The MetOp data we used are the Interim Climate
Data Records (ICDRs). It provides reprocessed bending angle, refractivity, dry temperature, 1dVar products. There products
have a high degree of internal consistency, which is important for climate monitoring.

Figure 1 shows the number of observations of all four types of observations in August. It can be seen that Fengyun
observations are generally of similar numbers to MetOp with observed profiles vary from 500 to 600 per day. However,
from 18 August to 24 August, the numbers FY3D and FY3E-GPS decrease with values are about 400-500. Panel (b) shows
the latitudinal distributions of these four types of RO observations during three days of Aug 14-16, 2023. RO observations
are found to be largest in two middle latitudes of both two hemispheres, i.e., 20° S -60° S and 20° N - 60°N. In the two polar
regions, observations are smallest with numbers are about 200 for the three days. Panel (c) shows numbers of setting
observations and rising observations. It can be seen the differences between rising and setting events for all four types of
observations are generally within 20%. From figure 1, we can see that either the number of RO observations, and also
latitudinal distributions or the types of RO events between Fengyun and MetOp satellites are overall of similar
characteristics.

In order to calculate RO biases and observations, we also used European Centre for Medium-Range Weather Forecasts
(ECMWF) reanalysis version 5 (ERA5) data in this study as a comparison to calculate these uncertainties. ERA5 is the fifth-
generation ECMWF atmospheric reanalysis of the global weather and climate (Hersbach et al., 2019, 2020; Simmons et al.,
2020). It was produced for the European Copernicus Climate Change (C3S) by ECMWF and replaces the ERA-Interim
reanalysis (Dee et al., 2011), which stopped being produced by August 2019. Such reanalysis data combines model data with
observations from across the world into a globally complete and consistent dataset using the laws of physics. It provides
atmospheric profiles on regular spatial grids and temporal layers. Most of the current researches used ERA5 data on 37
vertical levels. However this type of data is only up to about 40 km. In order to compare data from middle stratospheric
above, this research used 137 vertical levels' data from the surface up to an altitude of about 80 km. Four time layers, in



terms of 00, 06, 12 and 18 UTC are used. In order to match with the horizontal resolution of RO data, we used the 2.5° lat ×2.5° lon resolution ERA5 data.

## 2.2 Methodology

In designing a quality control scheme for RO bending angles, we reviewed several published quality control schemes. For instance, Scherllin-Pirscher et al. (2015) developed a series of quality control schemes for constructing BAROCLIM bending angle model. Angerer et al. (2017) proposed a new set of quality control schemes for RO bending angles. Additionally, ROMSAF has developed a series of quality control systems for its reprocessing system (Syndergaard et al., 2018). We used these quality control schemes as example, and tested the application of these existing quality control schemes to Fengyun

data and found that these schemes cannot fully meet the requirements of Fengyun data. For example, current schemes lack quality control in the height range of 30 – 50 km. However, we found that quality control in this height range are necessary since some Fengyun data may have outliers. Moreover, some previous schemes employ a minimum bending angle quality control method. However, it is also not suitable for MetOp data since MetOp exhibits lower noise level in its RO bending angles (Angerer et al., 2017).

Based on the above empirical quality control schemes and our own analysis on Fengyun data in this study, we designed an empirically derived bending angle quality control scheme (as shown in Table 1) for Fengyun and MetOp products. This scheme includes seven quality flags. Quality flags 1 to 3 (QF1 to QF3) are used to detect and reject outliers in bending angles: QF1, QF2, and QF3 are used to detect outliers in the height range of 50–80 km, 35–50 km, and 10–35 km height ranges, respectively. The thresholds are empirically obtained based on the experience of existing publications as discussed

above and also our internal tests. QF4 and QF5 are used to reject RO profiles with excessive bending angle noise at high-altitude regions. Bending angle biases and noises are calculated as the mean systematic differences and standard deviations of ionosphere-corrected bending angle against ERA5 bending angle from the height range of 65 km to 80 km (or to the top of a profile). The specifically equations can be seen from Pirscher et al., 2010. QF4 is used to reject profiles where the bending angle bias exceeds the noise level—indicating that bias-inducing error sources are so significant that they outweigh

the noise, making such bending angles unreliable. Furthermore, QF5 is used to reject profiles where bending angle noise exceeds 22 µrad. It is believed that if one of the bending angle quality flags is not equal to zero (good quality), this profile is not reliable and should be rejected.

In addition to bending angle quality, we also check quality of refractivity and temperature profiles in the height range of 10 – 35 km height range. This is used to further reject RO events that may suffer from large errors. Both refractivity and

temperature profiles are checked in the height range of 10-35 km. If refractivity difference profile exceed threshold of 10% (QF6) or if a temperature profile exceed threshold of 10 K (QF7), then this profile is discarded. A QF0 is also used




representing that this profile is of good quality that none of the quality issues are detected. A QF8 is used representing that at least one of the seven quality flags are not equal to zero.

## 3 Ionosphere-corrected bending angle quality

Based on the above methodology, this section presents the Fengyun bending angle quality evaluation results. Subsection 3.1 presents bending angle statistical errors as an overall view for understanding Fengyun bending angle quality. Subsection 3.2 introduces bending angle outlier rejection results and the characteristics of these quality flags are also analysed. Subsection 3.3 then introduces the quality issues related to biases and noises and their characteristics.

### 3.1 Bending angle statistical errors

Figure 2 presents systematic differences and standard deviations between ionosphere-corrected bending angles and ERA5 bending angles across six latitudinal bands globally: the entire globe (90°S to 90°N, Global), low latitudes (20°S to 20°N, Tropics = TRO), mid-latitudes (20°S/N to 60°S/N, with SHSM/NHSM referring to the southern/northern hemisphere subtropics and mid-latitudes), and high latitudes (60°S/N to 90°S/N, SHP/NHP = southern/northern hemisphere high latitudes). These profiles are derived from QF=0 data to avoid the influences of large biases and noises. The systematic differences among all four observations below 40 km are small and comparable. However, discrepancies become apparent above 40 km. The Fengyun series exhibit larger biases, while MetOp has the smallest biases. In terms of bending angle noise, FY3E-GPS and FY3D (which also uses a GPS receiver) have the highest levels of bending angle noise—approximately 1% to 3% larger than those of MetOp. The bending angle noise of FY3E-BDS falls in the middle, being roughly 0.5% to 2% higher than that of MetOp. This suggests that the GNOS BDS RO receiver produces smaller noises than the GNOS GPS RO receiver.

When comparing statistical errors across different latitudinal bands, largest errors are found in the SHP (Southern Hemisphere Polar) region. This can be attributed to the high uncertainty of ECMWF data in polar regions (Li et al., 2013, 2015). A spike transition at 26 km is observable in the error profiles of Fengyun satellites. This is primarily due to the transition from the geometric optics method to the wave optics method during bending angle retrieval. These spikes are even more pronounced in polar and tropical regions. However, they have been reduced in the latest version of Fengyun data. In the calculation of statistical errors, we found that enhancing quality control in the middle and upper stratosphere helps to further filter out outliers of Fengyun satellites—particularly for FY3E-GPS. This is the reason we introduced QF=3: specifically, if the absolute difference between values at 35 km and 50 km exceeds 100%, the profile is discarded (see Section 2). This criterion removes approximately 2% of additional Fengyun profiles but is highly effective in eliminating outliers in the middle and upper stratosphere.



Figure 3 presents the exemplary correlation functions and correlation coefficient contours of ionosphere-corrected bending angles. The top two rows display correlation functions at exemplary heights of 20 km, 40 km, and 60 km, covering the same six latitudinal bands as shown in Figure 2. The third row shows the correlation coefficient contours for FY3D, FY3E-GPS, and MetOp. Since the correlation coefficients of FY3E-BDS are similar to those of FY3E-GPS, the contours for FY3E-BDS are not included here. Examining the first two rows, the ionosphere-corrected bending angle correlation coefficients of all three satellites are generally comparable at the main peak, with shapes resembling an exponential function and correlation lengths of approximately 1 km. The correlation functions exhibit greater noise in the TRO (Tropical) and NHP (Northern Hemisphere Polar) regions, which is attributed to the smaller number of atmospheric profiles available in these areas. All Fengyun bending angles show anomalous curves below 26 km; this phenomenon is also associated with the transition from the Geometric Optics (GO) method to the Wave Optics (WO) method. This finding is also consistent with Lewis et al., 2025. The third row presents the correlation coefficients of all three sets of observations. Overall, the correlations are weak and comparable across the three observations. Similarly, the correlation coefficients of Fengyun bending angles exhibit abnormal biases of 0.2–0.4 from 26 km downward.

## 3.2 Bending angle quality evaluation

Figure 4 presents the daily time series of the percentage of poor-quality flags for RO bending angles across all four observational datasets. To quantify the characteristics of each quality flag, individual profiles were inspected for all flag categories; consequently, a single profile may be assigned multiple quality flags. Focusing on QF8 (denoted by red dots), which indicates overall poor-quality profiles, FY3E-GPS profiles exhibit the highest percentage of poor-quality profiles among all datasets, ranging from 35% to 45%. FY3E-BDS bending angles rank second, with their proportion of poor-quality profiles also spanning 30% to 40%. The quality of FY3D observations is superior to that of FY3E: the percentage of poor-quality profiles for FY3D mostly varies between 20% and 30%. In contrast, MetOp data demonstrate the best quality, with the percentage of poor-quality profiles typically around 20%. Since the overall quality also include quality control on refractivity, which we set a strict criteria, the bad quality percentage are higher than that solely use bending angle.

An analysis of bending angle quality (QF1–QF5) reveals that MetOp bending angles have minimal quality issues. This can be attributed to the high-quality GRAS radio occultation (RO) receiver and the strict internal quality control system integrated into its ROPP software. Only a small fraction (approximately 4%) of MetOp profiles are flagged with QF4, which indicates quality problems where bending angle biases are larger than the associated noises. This low QF4 percentage is primarily due to the very high signal-to-noise ratio (SNR) of the MetOp RO receiver (Angerer et al., 2017). Therefore, QF4 might not be truly useful for MetOp observations. In contrast, Fengyun datasets—particularly FY3E—exhibit much more bending angle quality control issues at high altitudes. When comparing QF1 to QF3 (which assess bending angle outliers across three distinct height ranges), the percentages of these flags are generally comparable. This suggests that if a profile





exceeds the threshold in one height range, it is highly likely to exceed thresholds in other height ranges. However, the percentages of these QFs do not fully overlap, which emphasizes the necessity of evaluating bending angle quality across

multiple height ranges. For FY3D and FY3E-BDS, QF3 (which targets the 10–35 km height range) has the highest percentage. For FY3E-GPS, QF2 (focusing on the 35–50 km range) is more prominent, indicating a greater possibility for outliers in the middle and upper stratosphere. As noted in the analysis of bending angle statistical errors (section 3.1), removing this outlier effect in the 35–50 km range would result in larger biases for FY3E-GPS—a phenomenon not observed in the other three datasets.

Quality flags 4 and 5 (QF4 and QF5) are associated with bending angle biases and noises. For Fengyun bending angles, FY3D demonstrates overall smaller bending angle biases and noise quality problems compared to the two FY3E bending angles. Their percentage of QF4 vary around 10%, while the percentage of QF5 is consistently less than 5%. For FY3E-BDS, QF4 percentages are generally small (less than 5%), whereas QF5 (large bending angle noises) percentages range from 15% to 20%. This indicates that FY3E-BDS bending angles are prone to significant noise-related quality issues. For FY3E-GPS

bending angles, both QF4 and QF5 percentages range from 10% to 15%, suggesting that this dataset is affected by both large biases and noise-related quality problems. As discussed earlier, only approximately 4% of METC bending angle profiles exhibit biases larger than noise (QF4).

Quality flags 6 and 7 represent the quality of refractivity and temperature, respectively. Since our quality control for refractivity is relatively strict, QF6 has the highest percentage among all quality flags. Again, MetOp refractivity has the

smallest percentage, at around 15%, followed by FY3D refractivity, which is approximately 20%. The two FY3E datasets have the highest percentages, at around 30%. For QF7, which represents temperature quality, the percentage for MetOp is nearly zero. The percentages for FY3D and FY3E-GPS are also small, at less than 5%. However, FY3E-BDS shows approximately 10% of temperature data with poor quality.

The bottom panel of Figure 4 shows the monthly mean percentage of each quality flag. Focusing on the overall quality of

bending angle profiles (QF1–QF3), MetOp observations exhibit almost no quality issues. The proportion of quality problems for FY3D is consistently below 10%, which is significantly lower than the approximately 20% observed for both FY3E-GPS and FY3E-BDS. Regarding QF4, FY3E-GPS has notably higher percentages (exceeding 10%) compared to the other three datasets, whose QF4 percentages all remain below 5%. Such biases of FY3E-GPS bending angles will be further illustrated in section 3.3. For QF5, FY3E-BDS has the highest percentage (around 15%), followed by FY3E-GPS and then FY3D;

MetOp data show almost no QF5-related quality issues. For QF6, both FY3E datasets (FY3E-GPS and FY3E-BDS) have large percentages (close to 30%), followed by FY3D (approximately 20%) and MetOp (around 15%). For QF7, FY3E-BDS has the highest percentage (about 10%), while FY3D and FY3E-GPS have QF7 percentages of less than 5%; MetOp data show no temperature-related quality issues. For QF8 (which represents overall profile quality), approximately 40% of FY3E-GPS observations are classified as poor quality, followed by FY3E-BDS (35%), FY3D (25%), and MetOp (less than 20%).





Figure 5 shows monthly mean percentage of each quality flags in the same six altitude bands as Figure 2. It can be seen that for almost all quality flags, the percentage of bad quality flags are most distinctive in TRO (20°S-20°N) and NHSM (20°N-60°N) regions. This can be attributed to that these two regions suffer more from the disturbance of ionospheric residual errors and also the influences of water vapour. Percentage of bad quality flags ranks the second in NHP region (northern hemisphere polar), especially for FY3E-BDS observations. The reason of this needs further investigations.

**3.3 Bending angle biases and noises**

Figures 6 to 8 show the characteristics of bending angle biases and noises for the four types of RO bending angles. These results are obtained using all QF=0 RO profiles. Figure 6 presents the individual RO bending angle biases (panel (a)) and noises (panel (b)) over three exemplary days: 14–16 August 2023. The individual bending angle biases of MetOp are overall small, with values varying within ±1 μrad. In contrast, the biases of Fengyun are larger than those of MetOp data. The noises

of MetOp are mostly within 2 μrad. However, the bending angle noises of Fengyun are significantly larger—many of these noises range in magnitude from 5 to 15 μrad. Panels (c) and (d) show the percentages of bending angle biases and noises that fall into each error range, respectively. MetOp data exhibit the smallest bending angle biases and noises, while Fengyun observations show generally larger biases and noises. Among the Fengyun series, FY3E-GPS exhibits larger biases than the other two Fengyun observations, and FY3D-BDS exhibits smaller bending angle noises than the other two Fengyun

observations. A more comprehensive overview of the monthly results is presented in Figure 8.

Figure 7 illustrates the percentages of bending angle biases and noises within the same ranges as Figure 6, but separated by rising and setting events. Darker colours represent positive biases, while lighter colours represent negative biases. For MetOp data, the percentages of rising and setting events are overall similar, with differences of less than 2%. For Fengyun data, however, the differences between setting and rising events are more pronounced. In the case of FY3D, rising events

exhibit generally smaller biases than setting events, though these overall differences are slight. For FY3E-GPS, rising events show significantly larger bias values than setting events, and most of the large biases are negative. Within the bias range of 0.5–1.0 μrad, the percentage of rising events is 30% higher than that of setting events. For FY3E-BDS, the biases of rising events are generally smaller: the percentage of rising events with biases falling into the smallest range (0–0.2 μrad) is 30% higher than that of setting RO events.

Figure 8 presents the temporal series of daily mean biases and noises (upper panel) and the percentages of biases and noises falling within specific error ranges (bottom two panels). Focusing on the upper panel, both the RO bending angle biases and noises exhibit no significant temporal variations. MetOp shows the smallest bending angle biases, with values around -0.5 μrad. FY3D and FY3E-BDS rank the second, with values around -0.1 μrad. In contrast, FY3E-GPS exhibits significantly larger bending angle biases than the other three datasets, with values varying around -0.4 μrad. This result is consistent with

the findings of the ROMSAF report (Lewis, et al., 2025). Both our results and the ROMSAF report found large bending



angle biases in FY3E observations. We further found that such large biases are mainly comes from the FY3E-GPS observations rather than FY3E-BDS observations. In addition, the ROMSAF report suggest a positive biases while we found negative biases. This may due to different background data are used for the calculation of biases and noises at high altitudes. Turning to bending angle noises, MetOp again has the smallest values, which vary around 1 µrad. FY3E-BDS ranks second,

with values varying around 1.5 µrad. FY3D and FY3E-GPS—both equipped with GNOS-GPS receivers—exhibit similar noise values, approximately 2.5 µrad.

A further examination of the bias and noise ranges (bottom two panels of Figure 8) shows that more than 90% of MetOp bending angle biases fall within the ±0.5 µrad range, with more than 50% within the ±0.2 µrad range. More than 80% of the bending angle biases for FY3E-BDS and FY3D also fall within ranges smaller than ±0.5 µrad. For FY3E-GPS biases,

however, only approximately 60% fall within ranges less than 0.5 µrad, while the remaining values fall within ranges larger than 0.5 µrad. This further explains why the daily mean bending angle biases of FY3E-GPS are much larger than those of the other three datasets (Panel (a)). Turning to RO bending angle noises, approximately 90% of MetOp noises fall within the 0–2 µrad range, followed by FY3E-BDS, for which about 80% of the noises fall within this range. For FY3D and FY3E-GPS, only about 40% of the noises fall within this range. More than 40% of the bending angle noises for FY3D and FY3E-GPS

fall within the 2–4 µrad range, and the remaining 20% fall within ranges larger than 4 µrad. This further explains why the observations from FY3D and FY3E-GPS exhibit larger noises than those from the other two datasets.

## 4 Optimized bending angle, refractivity and temperature

Figure 9 presents the systematic differences and standard deviations of statistically optimized bending angles across six latitude bands (upper two panels) and their correlation coefficients (third panel). When examining the systematic differences

overall, the Fengyun optimized bending angles exhibit error magnitudes similar to those of MetOp at high altitudes (above 50 km). This contrasts with the error characteristics of ionosphere-corrected bending angles. While the standard deviations of Fengyun's optimized bending angles below 50 km remain larger than those of MetOp data, the discrepancies are smaller than those of ionosphere-corrected bending angles. Above 50 km, however, the standard deviations of MetOp data are larger than those of Fengyun data—this also contrasts with the case of ionosphere-corrected bending angles. These results suggest

that the optimized bending angles from Fengyun satellites are more strongly corrected by background data. This conclusion is further supported by the bottom panel, which shows the correlation coefficients. Focusing on the correlation coefficients of Fengyun's statistically optimized bending angles, strong correlations are observed at high altitudes. If strong correlations are detected, it indicates that bending angles are strongly influenced by one data sources. This further confirms that Fengyun's statistically optimized bending angles are influenced by background bending angles at high altitudes. In contrast,



the correlation coefficients of MetOp optimized bending angle do not exhibit such strong correlations at high altitudes; their values are similar to those of ionosphere-corrected bending angles (see Figure 3).

To further understand the relationship between ionosphere-corrected bending angles and statistically optimized bending angles, Figure 10 presents the systematic differences and standard deviations of ionosphere-corrected bending angles relative to optimized bending angles. On a global scale, MetOp's ionosphere-corrected bending angles exhibit overall small

differences (less than 0.5 K) compared to its optimized bending angles up to an altitude of 60 km. In contrast, for Fengyun data, such small differences—with magnitudes similar to those of MetOp—are only observed below 50 km, an altitude 10 km lower than that of MetOp. Above 60 km, the differences in MetOp data gradually increase. However, below 70 km, these values remain mostly within ±2 K. This indicates that MetOp's ionosphere-corrected bending angles remain close to their optimized counterparts even up to 70 km. For Fengyun data, by comparison, the differences rise significantly above 50 km.

This suggests that Fengyun's optimized bending angles are more strongly influenced by background bending angles at altitudes above 50 km. Turning to standard deviations: those of MetOp are less than 0.5% below 50 km. In contrast, the standard deviations of Fengyun data increase sharply from 30 km upward. This further confirms that Fengyun's optimized bending angles are heavily affected by background data at altitudes above 50 km.

Figure 11 presents the systematic differences in refractivity and temperature across the same six altitude bands as Figure 6. It

can be observed that MetOp exhibits the smallest refractivity errors in the SHSM, SHP, and TRO regions. In the NHSM and NHP regions, MetOp data are consistent with Fengyun observations below 40 km. Above 40 km, however, Fengyun shows smaller refractivity errors. This is mainly because the differences in the optimized bending angles of Fengyun data are smaller than those of MetOp data above 50 km in these two regions. Consequently, refractivity, which is retrieved from optimized bending angle, show similar characteristics. A similar situation is found for temperature. Below 30 km, the

systematic differences in temperature among all four types of RO observations are generally similar. Above 30 km, MetOp exhibits the smallest temperature systematic differences in the SHSM, SHP, and TRO regions. In the NHSM and NHP regions, by contrast, Fengyun observations show smaller differences. These results are consistent with those of the optimized bending angles and refractivity. This is also attributed to the uncertainty propagation of the optimized bending angles (Li et al., 2019).

**5 Conclusion**

This study systematically assesses the quality of ionosphere-corrected bending angles of Fengyun D and E (both GPS and BDS) satellites using ERA5 data as references, and with MetOp data as comparisons. An empirical quality control scheme is developed in this study. The ionosphere-corrected bending angles are evaluated in a twofold way. First, they are checked from three height rages covering from 10 to 80 km. If bending angles exceed empirically determined thresholds, the profile

will be rejected and corresponding quality flag would be assigned. Secondly, the biases and noises of ionosphere-corrected



bending angles are calculated and if they did not pass our quality control, the profile is also rejected and quality flag would be assigned. The quality control scheme and thresholds are determined based on experiences of existing researches and our empirically analysis of Fengyun and MetOp bending angles.

A comparison of RO ionosphere-corrected bending angles with ERA5 bending angles shows that Fengyun's ionosphere-corrected bending angles are generally consistent with MetOp's bending angles below 40 km. Above 40 km, the statistical errors of Fengyun's bending angles are consistently larger than that of MetOp's bending angle. FY3E-GPS and FY3D exhibit the largest bending angle noises, which are approximately 1%–3% greater than those of MetOp. FY3E-BDS's bending angle noises fall in between, being roughly 0.5%–2% greater than MetOp's—this suggests that GNOS BDS RO receivers produce smaller noises than GNOS GPS RO receivers. The bending angle correlations of all three Fengyun

observations are generally consistent above 30 km.

Focusing on the first set of quality flags (QF1–QF3)—which are used to identify outliers in bending angle profiles—we note that the percentage of poor-quality profiles for MetOp is nearly zero. This is attributed to its high-performance receiver and strict internal quality control system. FY3D ranks second, with fewer than 10% of its profiles containing outliers. The two bending angle datasets from FY3E (from GPS and BDS receivers) have the highest percentages of poor bending angle

quality profiles, with values around 20%. Turning to quality issues related to bending angle biases and noises, we also note that MetOp and FY3D have few such issues (less than 5%). However, the percentage of large biases of FY3E-GPS are approximately to 15% and the percentage of large noises of FY3E-BDs are also approximately to 15%.

Further check the magnitudes of bending angle biases and noises (data with no quality issues) show that MetOp bending angle biases are smallest with biases are around -0.5 μrad. FY3D and FY3E-BDS bending angle biases rank the second with

biases are around -0.1 μrad. FY3E-GPS bending angles show distinctive much larger biases than the other three observations with values varying around -0.4 μrad, and most of these large biases are detected in rising events. Focusing on bending angle noises, MetOp bending angle biases again are smallest with values varying around 1μrad. FY3E-BDS bending angles rank the second, with values varying around 1.5 μrad. FY3D and FY3E-GPS, which are all GNOS-GPS receivers, show similar noises values which are about 2.5 μrad.

The statistical errors of optimized bending angles are also calculated. Both systematic differences and standard deviations of Fengyun bending angles at high altitudes are significantly reduced compared to the ionosphere-corrected bending angle. Furthermore, Fengyun optimized bending angle reveal larger correlations above 50 km. This all suggest that the optimized bending angle are strongly affected by the background bending angle at high altitudes. Statistical errors of refractivity and temperature are also calculated. Below 40 km, refractivity errors of all four observations are overall consistent. However,

above 40 km, MetOp refractivity still outperforms in SHSM, SHP, TRO regions. In the NHSM and NHP regions, Fengyun refractivity outperforms. This may be caused by the strong weighting were given to the optimized bending angle in these two regions. Similar situation were found in temperature statistical errors.

In conclusion, the ionosphere-corrected bending angles from Fengyun RO satellites exhibit promising quality below 40 km. Above 40 km, however, both the biases and noises of Fengyun's bending angles are larger than those of MetOp data.



Notably, FY3E-GPS bending angles exhibit distinct negative biases in rising events. The sources of these biases and noises may include three factors: orbit determination errors, clock errors, or receiver noises/biases. From a review of current literature, however, we found that Fengyun data already achieve highly accurate orbit determination and clock performance. Therefore, residual biases and noises should be the primary cause of the large errors in Fengyun's bending angles at high altitudes. Our future work plan includes further investigating and quantifying the main causes of the large bending angle

errors in Fengyun data at high altitudes. We also aim to develop an empirical observation error model for the statistical optimization of Fengyun's ionosphere-corrected bending angles. It is expected that this work will further enhance the performance of Fengyun data in numerical weather prediction and climate studies, particularly for stratospheric applications.

*Code availability.* The code used to produce the results of this study is available from the corresponding author upon

qualified request.

*Data availability.* The (numeric) data underlying the results of this study are available from the corresponding author upon qualified request.

*Author contributions.* Ying Li implemented the new method, performed the analysis, produced most of the figures, and wrote the initial draft of the manuscript. Yan Liu served as primary co-author, providing advice and guidance on algorithm design, analysis, and figure production, and contributed to the writing of the manuscript. Wenwu Ding supported on the algorithm design and test, data analysis and revision of the paper manuscript. Mi Liao works on the algorithm design, data collection and analysis. Xingliang Huo contributed on the data analysis, validation and revision of the text of the paper.

Jinying Ye works on the data collection, validation and also part of the wring of the paper.

Acknowledgements. We acknowledge ECMWF (Reading, UK) for providing access to their analysis and forecast data. We also acknowledge ROMSAF for providing access to their MetOp data.

*Financial support.* The research at APM (Wuhan, China) was supported by the National Key Research & Development Program (No. 2023YFA1009100) and National Natural Science Foundation of China (Grant no. 42474063). At CMA, this work was supported by the Fengyun-3 03 batch satellite project "numerical forecast assimilation application of fengyun-3 03 batch ground application system (No.FY-3(03)-AS-8).

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

**Table 1.** Quality control flags of RO atmospheric profiles

| Quality Flags | Description |
|---|---|
| QF=0 | All quality control passed |
| QF=1 | Remove all profiles with bending angles larger than +40 μrad or smaller than -40 μrad between 50 and 80 km height |
| QF=2 | If absolute differences between ionosphere-corrected bending angle and ERA5 bending angles are larger than 100% in the height range of 35-50 km |
| QF=3 | If relative difference between ionosphere-corrected bending angle and ERA5 bending angles are larger than 20% between 10 and 35 km. |
| QF=4 | If RO bending angle bias is larger than its noise |
| QF=5 | If RO bending angle noise is larger than 22 μrad |
| QF=6 | If relative refractivity difference is larger than 10% during the height range from 10-35 km |
| QF=7 | If temperature difference is larger than 10 K during the height range from 10-35 km |
| QF=8 | If any of the above quality control is not passed |




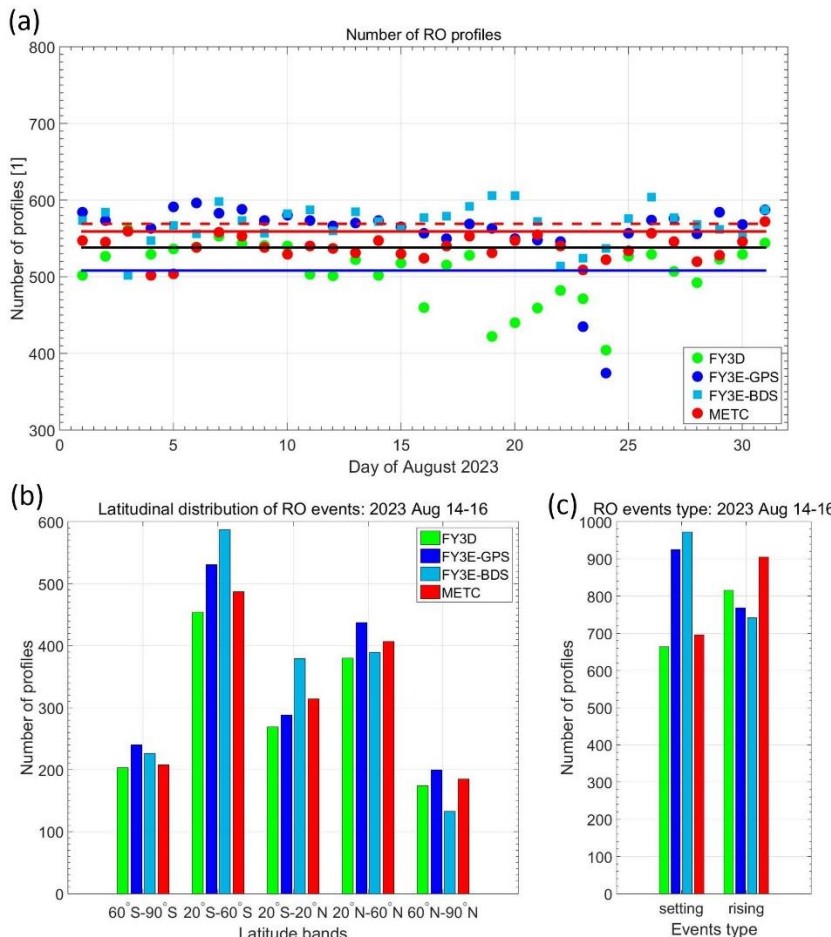

**Figure 1**: Panel (a): Daily number of FY3D, FY3E-GPS, FY3E-BDS events in August 2023; Panel (b): number of RO events during the three days of 14-16 August, 2023 in five latitude bands of SHP (60°S-90°S), SHSM(20°S-60°S), TRO(20°S-20°N), NHSM(20°N-60°N) and NHP(60°N-90°N); Panel (c): number of setting and rising events of the four types of RO observations during 14-16 August, 2023.





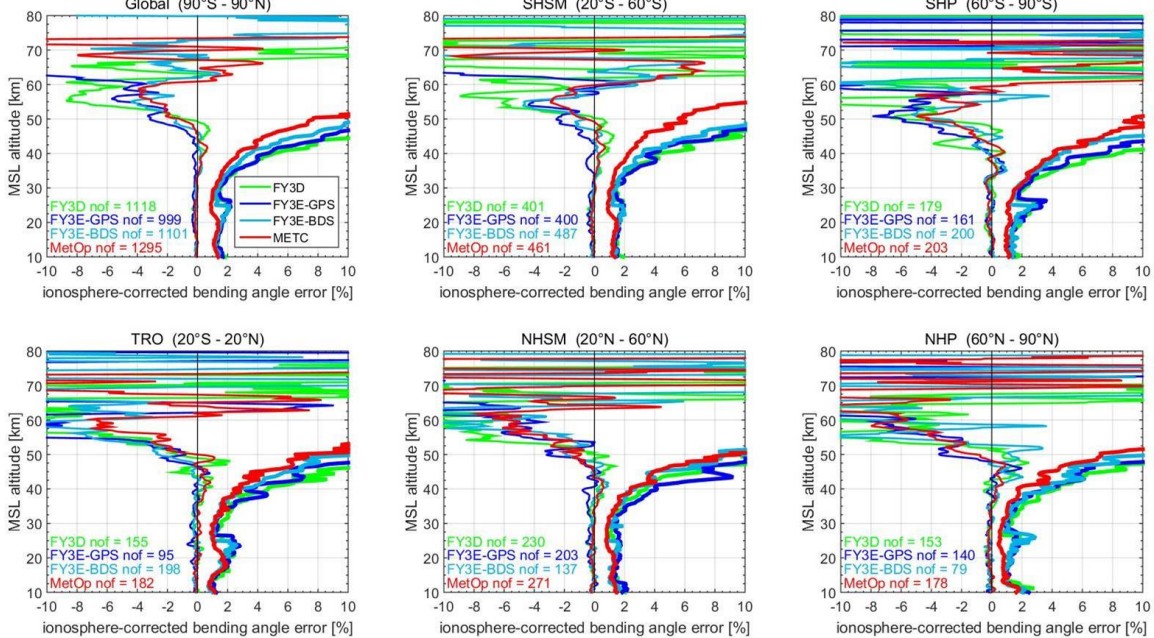

**Figure 2.** Upper two rows: systematic differences (thin lines) and standard deviations (thick lines) of ionosphere-corrected bending angle against ERA5 data during 14-16 August 2023 time period in six latitudinal bands of Global (90°S-90°N), SHP (60°S-90°S), SHSM (20°S-60°S), TRO (20°S-20°N), NHSM (20°N-60°N) and NHP (60°N-90°N), results of FY3D, FY3E-GPS, FY3E-BDS and MetOp are shown. Numbers of profiles in each region (e.g., FY3D-nof) are indicated in each panel.



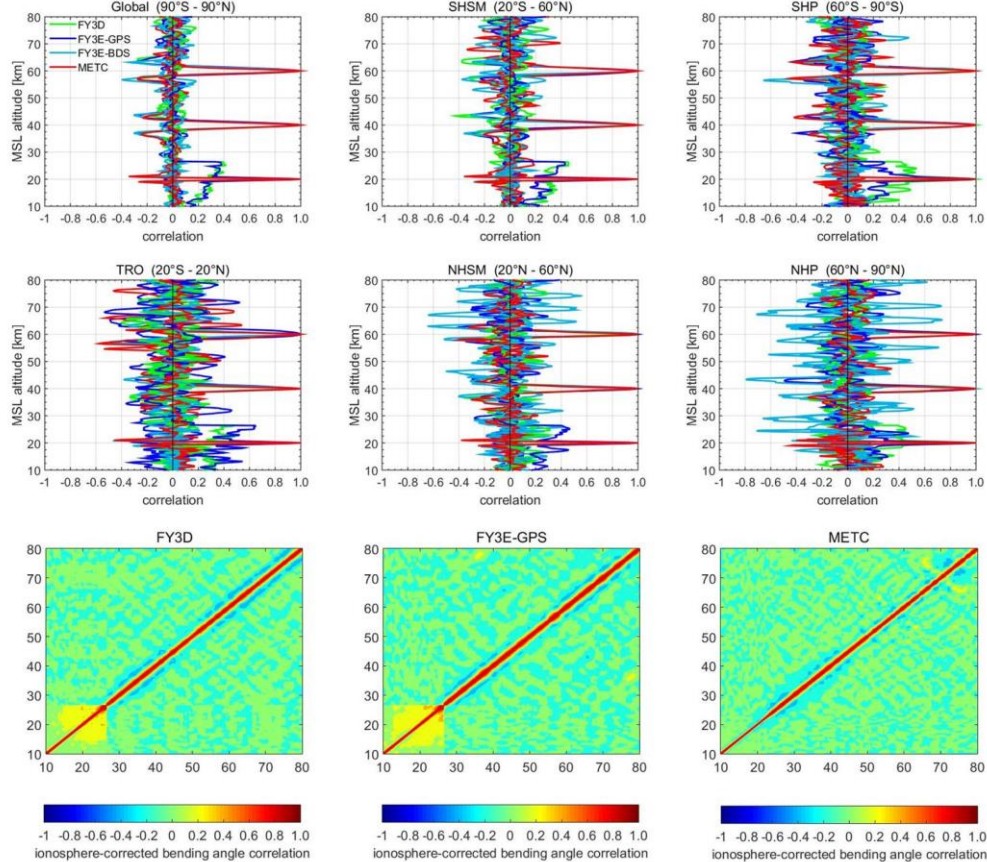

**Figure 3** Upper two rows: correlation functions of ionosphere-corrected bending angle at three exemplary heights of 20, 40 and 60 km across the same six latitudinal bands as Figure 2, and results of all four types of observations are shown. Third row: correlation coefficients of ionosphere-corrected bending angle of FY3D, FY3E-GPS and METC observations;





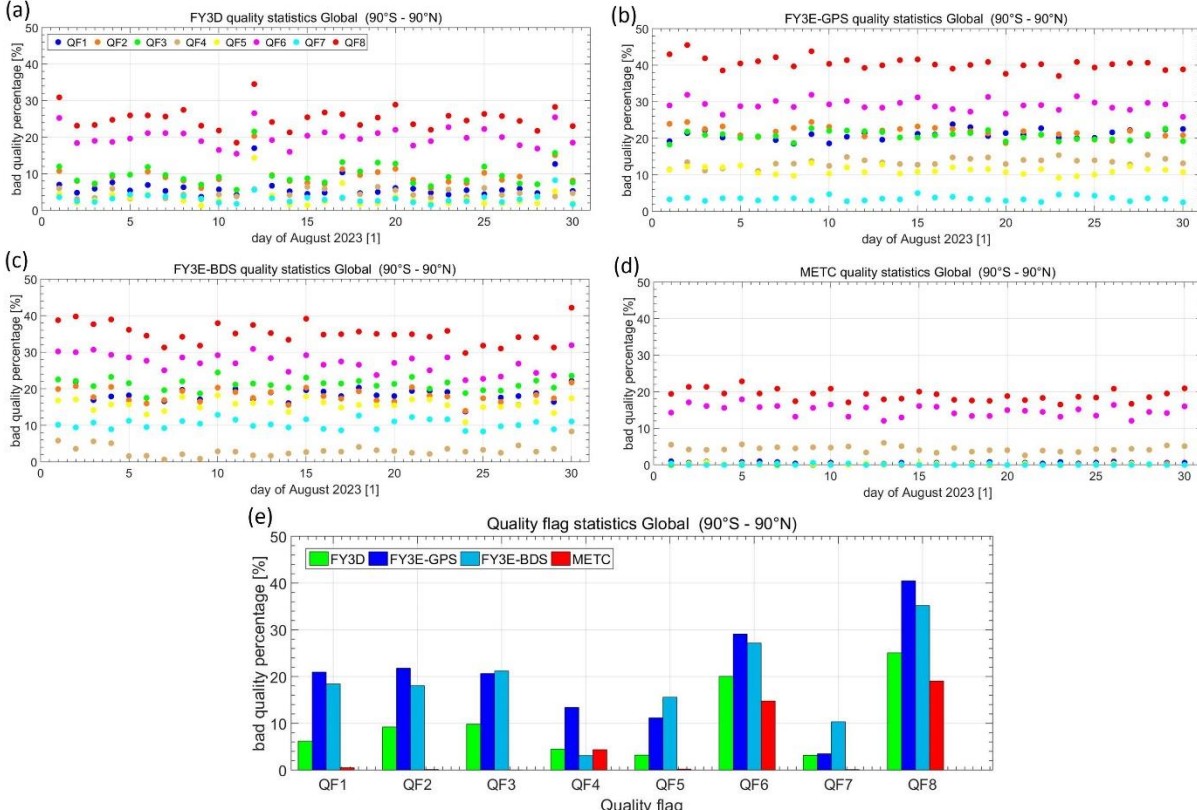

**Figure 4.** Panels (a) – (d): time series of daily mean percentage of bad quality flags (QF1-QF8 see table 1) for FY3D, FY3E-GPS, FY3E-BDS and METC observations, respectively. Panel (e): proportion comparison of the eight bad quality flags for each type of RO observations of August, 2023.







**Figure 5.** Percentage of bad quality flags in the same six latitude bands as Figure 2. Results are obtained using data from August, 2023.





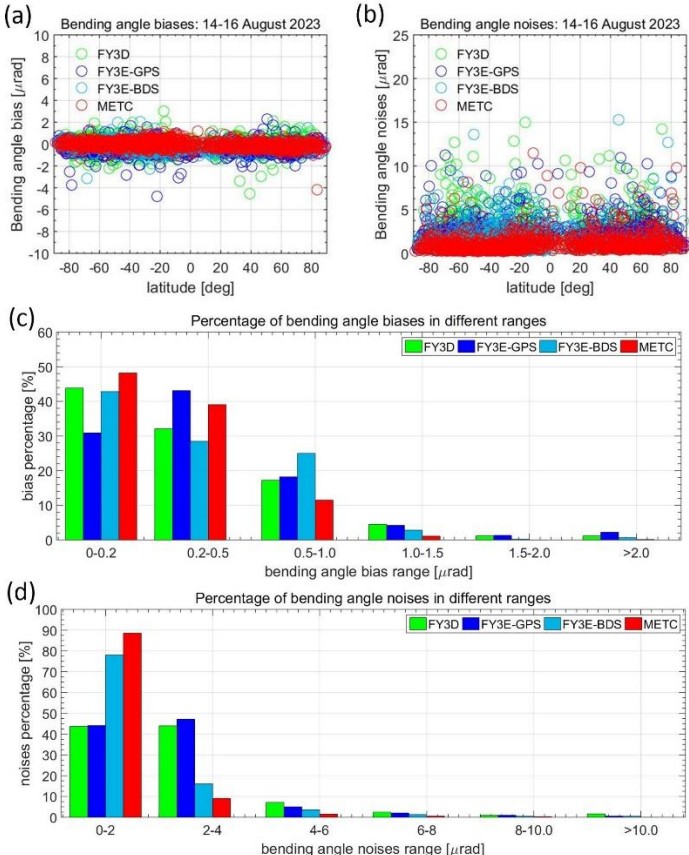

**Figure 6.** Panels (a) and (b): Latitudinal variations in RO bending angle biases and noises for three exemplary days (14–16 August); Panel (c): Percentages of biases falling within different value ranges; Panel (d): Percentages of RO bending angle noises falling within different value ranges.



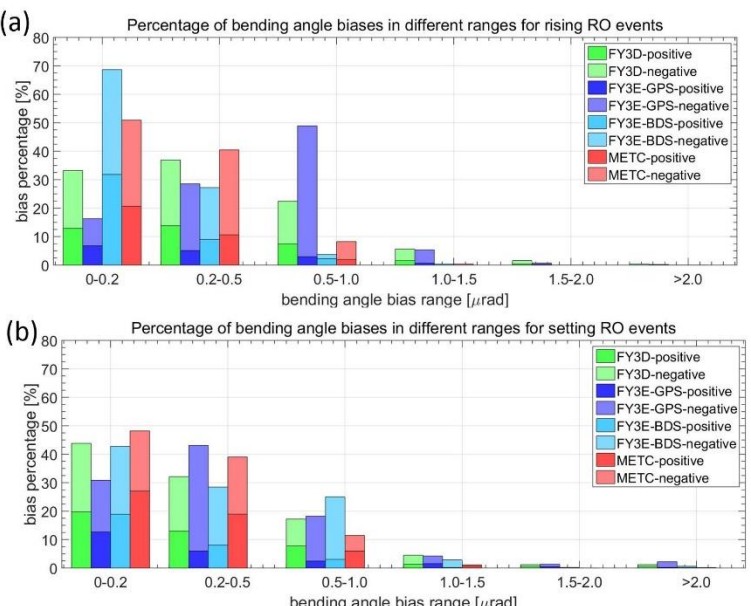

**Figure 7** Percentage of bending angle biases across six error ranges for rising (upper) and setting (bottom) events, and percentages values are obtained using data from 14-16 August, 2023.



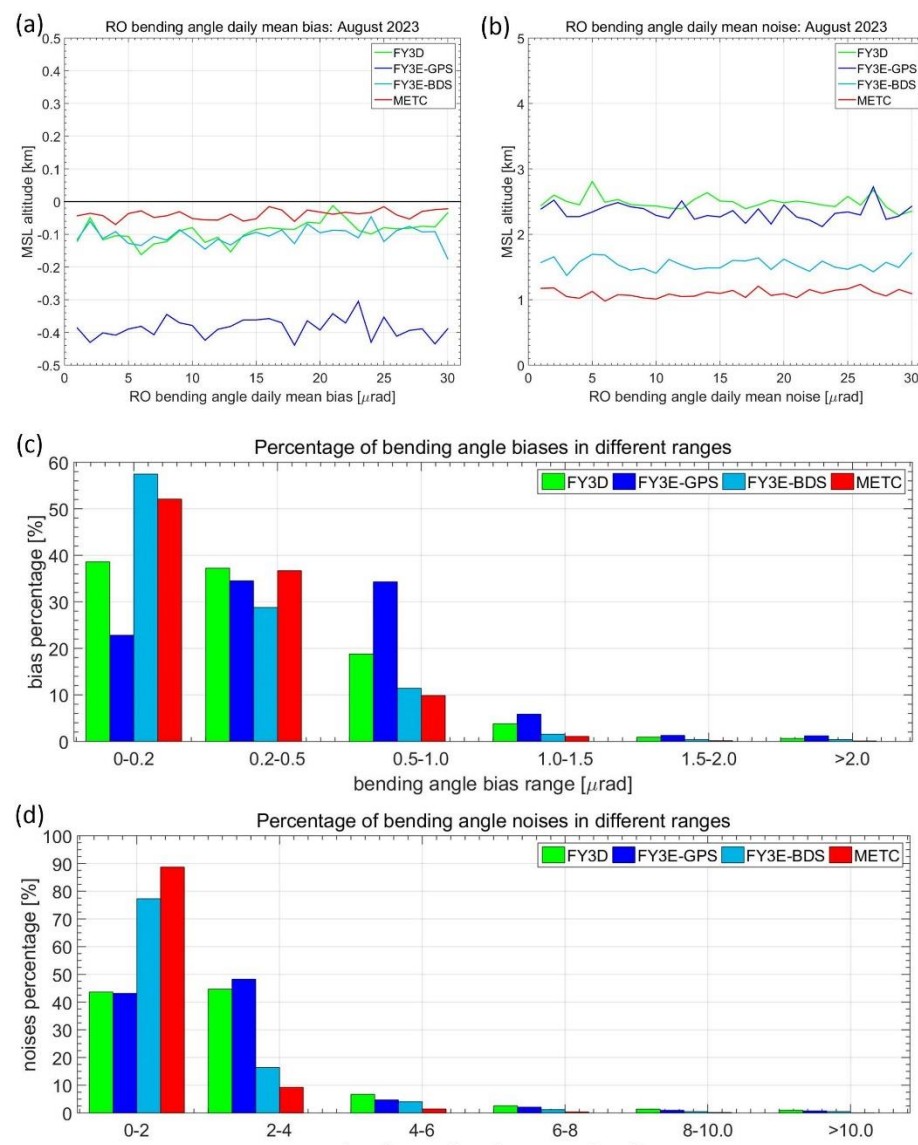


**Figure 8.** Panels (a) and (b): time series of daily mean bending angle biases (a) and noises (b) for August 2023; Panels (c) and (d): percentage of bending angle biases (c) and noises (d) across six error ranges.



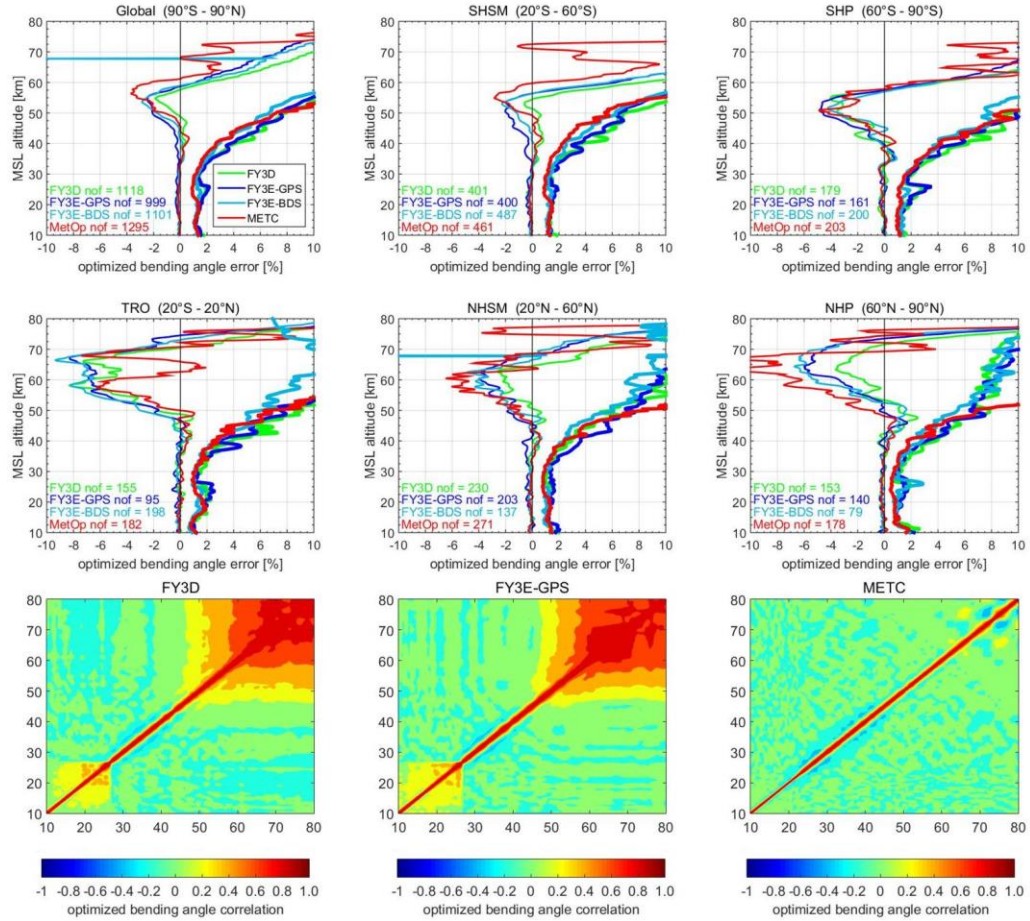

**Figure 9.** Upper two rows: systematic differences (thin lines) and standard deviations (thick lines) of optimized bending angle against ERA5 data during 14-16 August 2023 time period in the same six latitudinal bands as Figure 2. Bottom row: correlation coefficients of optimized bending angle of FY3D, FY3E-GPS and METC observations.



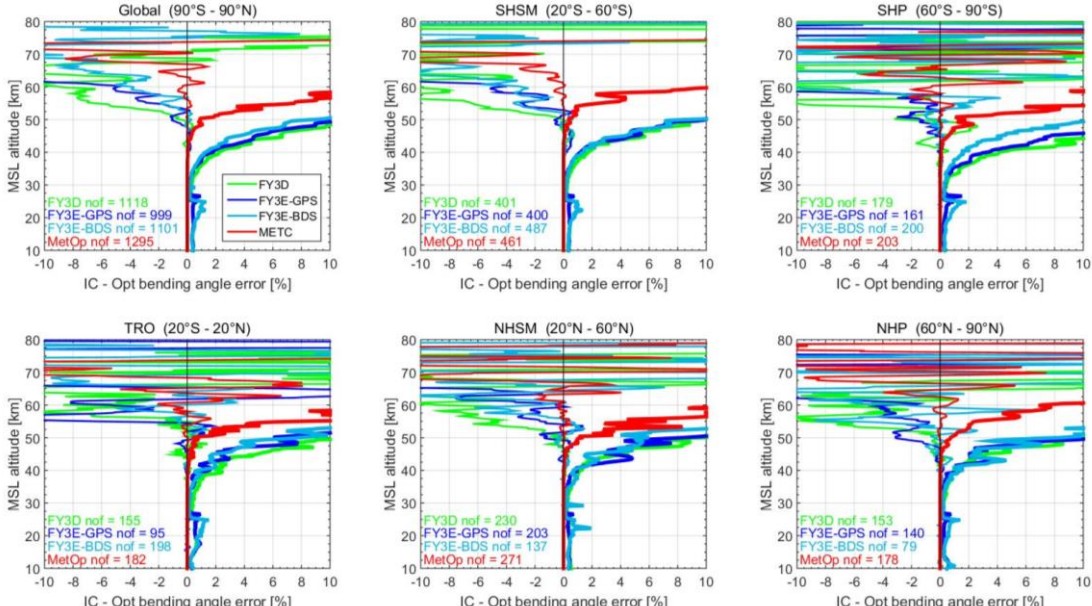

**Figure 10.** Systematic differences (thin lines) and standard deviations (thick lines) of ionosphere-corrected bending angle against optimized bending angle during 14-16 August 2023 time period in the same six latitudinal bands as Figure 2.



**Figure 11.** Systematic differences (thin lines) and standard deviations (thick lines) of refractivity (upper two rows) and temperature (bottom two row) against ERA5 data during 14-16 August 2023 time period in the same six latitudinal bands as Figure 2.