# Peer review of "Quality aspects of Fengyun3 D/E radio occultation bending angle products"

_EGUsphere, 2025_

## Author Comment (AC1)

**Responses to Reviewer#1**

**Overall comments:** the authors present a thorough investigation of the RO data quality of the FY3 satellites, using as reference ERA5 data and comparing it to a high-quality RO receiver (Metop-C). The quality is assessed using both ionospheric-corrected and statistically optimized bending angles. Comparisons are also performed for derived refractivity and temperature profiles. The results are useful and they could benefit from some changes and more information on a number of topics, listed below under "Main comments". Included are also minor comments and a list of language suggestions.

**Overall responses:** Thanks for the comments from Reviewer#1. We have revised the manuscript according to the major comments and minor comments. Please see the revised manuscript and our responses as below.

**Main comments:**

**Major comment 1:** Consider adding a description of the FY3 mission (e.g., orbital configurations, main features of the different satellite buses, description of the RO antennas, etc.). Having this information would make it easier to assess statements like the one on Ln 180-181 ("*This suggests that the GNOS BDS RO receiver produces smaller noises than the GNOS GPS RO receiver*") Based on this sentence, it looks like the BDS receiver is different from the GPS receiver. Is it so, or is it a GPS-only VS a GPS-and-BDS receiver?

**Response to major comment 1:** Thanks for this comment. After a detail literature review, internal discussion and consulting to relevant experts from NSSC, we can now conclude the following results:

1) FY3C and FY3D carry GNOS-I receiver, which is able to track GPS signals.

2) FY3E, FY3G and FY3G carried GNOS-II RO receivers, which is able to track both GPS and BDS signals.

3) The GNOS-II receiver is an integrated receiver capable of tracking both GPS and BDS signals. Our previous distinction between the GNOS BDS RO receiver and GNOS GPS RO receiver was inaccurate. In the revised manuscript, we have updated the relevant descriptions to refer to GNOS GPS observations and GNOS BDS observations.

4) The reason that GNOS GPS observations have larger noises than the BDS observations are still unknown. However, we have thought of two possible reasons. First, the constellation geometry of BDS, which includes GEO and IGSO satellites with much slower relative motion with respect to LEO platform. This leads to relatively smaller Doppler rate and more stable carrier-phase tracking for BDS occultation. Secondly, GPS satellites P2 signal operates in a code-free tracking model, which consequently results in a slightly inferior tracking performance.

In the updated manuscript, we have corrected corresponding statements, please see lines 65 to 66 in page 2 and lines 380 to 400 of pages 6 and 7 in the updated manuscript which are also shown below:

"*Among these series of satellites, the FY3C and FY3D carried the first version of GNOS receiver (GNOS-I) and FY3E to FY3G carried the second version of GNOS RO receiver (GNOS-II).*"

"*This suggests that the GNOS BDS observations have smaller noises than the GNOS GPS observations. This can be attributed to two reasons. First, the constellation geometry of BDS, which includes GEO and IGSO satellites with much slower relative motion with respect to LEO platform. This leads to relatively smaller Doppler rate and more stable carrier-phase tracking for BDS occultation. Second, GPS satellites P2 signal operates in a code-free tracking model, which consequently results in a slightly inferior tracking performance.*"

Other similar descriptions in the abstract, conclusions and other places have also been updated.

**Major comment 2:** Ln 186: It is mentioned that the spikes have been reduced in the latest version of the Fengyun data. This is very good, and some description of how this result has been achieved and possibly a figure illustrating it would be very nice.

**Response to major comment 2:** Thanks for this comment. After a detail literature review, internal discussion and consulting to relevant experts from CMA and NSSC, we can now conclude the following results (lines 403 to 407 of page 7 in the revised manuscript):

"*A spike transition at about 25 km is observable in the error profiles of Fengyun satellites. This is primarily due to two reasons: 1) extrapolation of L2 signal at fixed 25 km; 2) fixed transition height at 25 km). Since 2025, this algorithm has been updated. First, the extrapolation height of L2 will be determined by the quality of L2 signal. Second, the fixed transition height was changed from 25 km to 15 km. By this update, the standard deviations of refractivity has been reduced by about 0.5% in this height range.*"

Below the figure 1 of this document is the refractivity bias and RMSE before this update and figure 2 shows the improved refractivity bias and RMSE results.

[Figure]

Figure 1. Refractivity bias and RMSE results before revision

[Figure]

**Figure 2.** Refractivity bias and RMSE results after revision

**Major comment 3:** Figure 1, panel (c): It is rather unusual to have more rising than setting occultation. Can you double-check that the data is correct? If this is the case, please comment on this unexpected asymmetry.

**Response to major comment 3:** Thanks for pointing this out. Yes, we indeed find that in diagnosing RO setting and rising events, we made mistakes. The setting and rising event flags were inadvertently reversed in the data processing of FY3D, FY3E-BDS and METC RO observations. After the revision, FY3D, FY3E-GPS and METC all have more setting events than rising events. However, FY3E-BDS observations still have more rising events than setting events. We have discussed internally. However, we cannot quickly determine the reasons so far. This will be investigated in our next work.

In order to correct our mistakes, we have revised Figures 1 and 7 which is related to setting and rising events. The corresponding descriptions of these two figures are also updated. Please see lines 286 to 287 of page 4 and from lines 510 to 521 of page 10 in the revisied manuscript.

*"FY3D, FY3E-GPS and METC have more setting events than rising events. However, FY3E-BDS records more rising events, and the reason for this remains unknown, which requires further investigation in the future."*

*"Figure 7 illustrates the percentages of bending angle biases and noises within the same ranges as Figure 6, but separated by rising and setting events. Darker colours represent positive biases, while lighter colours represent negative biases. For setting events, the positive and negative biases of MetOp are of similar magnitudes overall. More than 85% of MetOp biases are less than 0.5 µrad. For the Fengyun satellite series, however, negative errors outnumber positive errors, and this trend is even more pronounced for FY3E-GPS bending angles. For example, within the bias range of 0.5–1.0 µrad, approximately 45% of the biases are negative, while only 5% are positive. When comparing the bias magnitudes across Fengyun satellites, FY3E-GPS bending angles show the largest biases, with about 50% of FY3E-GPS observations falling within the 0.5–1.0 µrad error range. Among all Fengyun satellites, FY3E-BDS has the*

*smallest biases, as over 95% of its biases are less than 0.5 μrad. FY3D biases fall in between, with roughly 70% of its biases below 0.5 μrad. For rising events (panel (b)), all four RO bending angles exhibit more negative biases than positive ones, and this characteristic is also more prominent for FY3E-GPS observations. When comparing setting and rising events, the proportion of large bending angle biases (greater than 0.5 μrad) in FY3E-GPS observations is less than 20%."*

**Major comment 4:** Ln 303: Please provide a description of how you obtain the statistically optimized results. Are you using the ROPP for this?

**Response to major comment 4:** Thanks. In fact, we use the statistical optimized results and subsequent refractivity and temperature results from the products of ROMSAF and CMA. Both centers use ROPP to process their statistical optimized results and subsequent retrieved profiles. In order to avoid this confusion, we have added a new description in section 2.2 and at the beginning of section 4 in the updated manuscript. Please see lines 271 to 274 of page 4 and lines 564 to 565 of page 11 in the updated manuscript (shown as below also):

*"In this study, we used Fengyun-3D (FY3D), Fengyun-3E GPS (FY3E-GPS), Fengyun-3E BDS (FY3E-BDS) and MetOp (METC) RO ionosphere-corrected bending angle observations, optimized bending angle, refractivity and dry temperature profiles."*

*"This section presents the statistical errors of optimized bending angle, refractivity and temperature. These profiles are directly obtained from CMA and ROMSAF (c.f., section 2.2). Both data centers use the ROPP software to process their RO retrievals."*

**Major comment 5:** Ln 381/382: You mention that the orbit determination (OD) and clock estimation of the FY3 is of very high quality. This is an important piece of information and it requires at least the references that support it.

**Response to major comment 5:** Thanks for this comment. We found the proof from following references. However, we cannot conclude that OD is not one of the reasons of the large biases and noises of FY3D (see also responses to major comment 6), we therefore delete the original statement.

Li et al., 2017 demonstrated that "FY-3C GNOS attains centimeter-level radial orbit precision and reliable clock estimation through integrated GPS and BDS dual-system observations, with 3D orbit accuracy reaching 2.7 cm via GPS-only POD and consistent clock performance supported by 30-s precise clock products from IGS and Wuhan University".

Li et al., 2018 reported that "FY-3C GNOS enables integrated orbit determination of BDS, GPS, and the satellite itself, delivering high-quality orbit and clock solutions with

centimeter-level radial precision, well-suited for meteorological and climate research applications."

Li, M., Li, W., Shi, C., Jiang, K., Guo, X., Dai, X., Meng, X., Yang, Z., Yang, G., and Liao, M.: Precise orbit determination of the Fengyun-3C satellite using onboard GPS and BDS observations, J. Geod., 91, 1313–1327, https://doi.org/10.1007/s00190-017-1027-9, 2017.

Li, X., Zhang, K., Zhang, Q., Zhang, W., Yuan, Y., and Li, X.: Integrated Orbit Determination of FengYun-3C, BDS, and GPS Satellites, Journal of Geophysical Research: Solid Earth, 123, 8143–8160, https://doi.org/10.1029/2018JB015481, 2018.

**Major comment 6:** To further support the statement that the OD is not the cause of the residual biases and noises, it would be very useful to see figures similar to, e.g., Fig. 2, but where the datasets are split by rising VS setting products. OD errors often show up as rising VS setting biases above ~ 50 km.

**Response to major comment 6:** After a detailed analysis, we found that our original statement on OD was not entirely accurate. We plotted the rising and setting events in a manner similar to Fig. 2 and observed that setting events above 50 km exhibited significantly larger negative biases than rising events. This result is consistent with our updated Figure 7. Furthermore, ROMSAF Report 47 also indicates that FY3E bending angle observations show discrepancies between setting and rising events. Therefore, we cannot conclude that OD has no influence on the retrieval profiles of Fengyun observations. We would like to thank you again for this comment, as it has helped improve our understanding of the impacts of OD errors on retrieved atmospheric profiles.

In the updated manuscript, We have revised relevant statements. Please see lines 651 to 658 of page 13 in the updated manuscript and also as below:

*"In conclusion, the ionosphere-corrected bending angles from Fengyun RO satellites exhibit promising quality below 40 km. Above 40 km, however, both the biases and noises of Fengyun's bending angles are larger than those of MetOp data. Notably, FY3E-GPS bending angles exhibit distinct negative biases. The sources of these biases and noises may include three factors: orbit determination errors, clock errors, or receiver noises/biases. Our future work plan includes further investigating and quantifying the main causes of the large bending angle errors in Fengyun data at high altitudes. We also aim to develop an empirical observation error model for the statistical optimization of Fengyun's ionosphere-corrected bending angles. It is expected that this work will further enhance the performance of Fengyun data in numerical weather prediction and climate studies, particularly for stratospheric applications."*

**Major comment 7:** Figures are not of pdf quality, so when zooming they become blurred.

**Response to major comment 7:** Thank you for this comment. In the revised manuscript, we have updated Figures 1, 7 and 8 with higher resolution. All figures now have adequate resolution and can be clearly visualized even when zoomed in.

**Major comment 8:** The y-labels of Figure 8 (panels (a) and (b)) are wrong.

**Response to major comment 8:** Thanks for this comment. We have corrected x and y labels of Figure 8 in the updated manuscript.

**Minor comments:**

**Minor comment 1:** Ln 71: Missing reference "Anthes, 2024". The correct one is this: https://journals.ametsoc.org/view/journals/bams/105/8/BAMS-D-23-0326.1.xml

**Response to minor comment 1:** Thank you for this reminder; we have added this reference to the reference list of the revised manuscript.

**Minor comment 2:** In several places the citation style is incorrect. E.g., in ln 79, "Liao et al., 2016b" should be "Liao et al. (2016b)"

**Response to minor comment 2:** Thanks for this comment. We have corrected these reference issues in the revised manuscript.

**Minor comment 3:** Paragraph starting on line 76: This paragraph presents a comprehensive list of literature. However, only for the paper by Liao et al. (2016b) a short description of the results is provided. Please report concise descriptions of the results of the other papers as well.

**Response to minor comment 3:** Thanks for this comment. We have updated corresponding literature description in the updated manuscript. Please see lines 95 to 103 in page 3 of the revised manuscript and also as below:

*"Sun et al. (2018) reviewed the FY3C GNOS instrument, RO data processing, and data quality evaluation. Their results further demonstrated that FY3C can provide accurate atmospheric profiles below 30 km and reliable ionospheric products. Bai et al. (2018) discussed the differences between the single-difference and zero-difference clock corrections applied to RO observations and found negligible discrepancies between the two methods. With the launch of the FY3E satellite, which is equipped with the GNOS-II receiver, several studies have further evaluated the quality of GNOS-II RO observations (Liu et al., 2023; He et al., 2013; Liang et al., 2024; Mo et al., 2024). These studies have further confirmed the high quality of GNOS-II RO profiles in the upper troposphere and lower stratosphere. Beyond these published papers, ROMSAF has also released two reports that analyse the data quality of FY3C and FY3E (Bowler et al., 2019; Lewis et al., 2025)."*

**Minor comment 4:** Ln 94: What is the NOMEX experiment? Do you mean the ROMEX? If so, please note that the data-collection part has already been completed (see reference in the comment to Ln 71 above).

**Response to minor comment 4:** Yes,we mean the ROMEX. Thanks for reminding us, we have deleted the information about ROMEX in this sentence.

**Minor comment 5:** Paragraph ln 135-144: Please provide a brief description of the schemes mentioned here. This would help the reader understand how your approach is more valid and/or relevant for FY3. If it makes no sense to do so because, e.g., they are all empirically based, provide a qualitative description of them.

**Response to minor comment 5:** Thanks for this comment, yes, our approach is all empirically based. We have revised corresponding description in the updated manuscript. Please refer to lines 318 to 325 in the revised manuscript and also as below:

"*The quality control schemes of our study are empirically based. First of all, we used the quality control of above mentioned studies as basis and then apply them to Fengyun and MetOp data. After that we compare the calculated statistical errors with results in previous literatures and then adjusting corresponding schemes. For example, current schemes lack quality control in the height range of 35 – 50 km. However, we found that quality control in this height range is necessary since some Fengyun data may have outliers. Moreover, some previous schemes employ a minimum bending angle quality control method. However, it is also not suitable for MetOp data since MetOp exhibits lower noise level in its RO bending angles (Angerer et al., 2017). By several iterative process, we finally obtained the quality control scheme in this study as shown in Table 1.*"

**Minor comment 6:** Ln 153: You are referring to a non-peer-reviewed publication. It's ok but at least do include the relevant equations.

**Response to minor comment 6:** OK, we included relevant equations in the updated manuscript, please see section 2.2.

**Minor comment 7:** Ln 157: I wouldn't use "It is believed" but something like "We assume that if one of the bending angles …".

**Response to minor comment 7:** Thanks, we have revised into "We assume that…",

**Minor comment 8:** Data availability: you indicate that FY3 data comes from the CMA reprocessing campaign and the Metop-C from the ICDRs. Is there an official way to access the CMA-reprocessed data? Could you indicate where the ICDRs are available?

**Response to minor comment 8:** Thanks for the comments. After a confirmation internally, we found that there are in fact no re-processed data of Fengyun series. CMA has updated some algorithms (as mentioned above for reducing the spikes between 20 to 30 km) since the beginning of 2025. However, for historical data, nothing has been changed. These data can be downloaded from Fengyun satellite data service of CMA, and the link is below:
https://data.nsmc.org.cn/DataPortal/cn/home/index.html

The ICDRs data are available at the website of the official ROMSAF website, and the link is as below:
https://rom-saf.eumetsat.int/product_archive.php

Sorry for the wrong description in our original manuscript. In revisied manuscript, we have revised corresponding description as below:

"*Fengyun data can be downloaded from Fengyun satellite data service provided by the National Satellite Meteorological Center (NSMC) CMA. It is also known that the Fengyun data has been updated since 2025 to improve its data quality below 30 km. However, this does not affect our results in this study since we focus on bending angle quality above 30 km. The MetOp data we used are the Interim Climate Data Records (ICDRs). Such data can be downloaded from the official website of ROMSAF.*"

We also revised the data availability section in the revised manuscript, please see also below:
"*Data availability. These data can be downloaded from Fengyun satellite data service of CMA, and the link is below: https://data.nsmc.org.cn/DataPortal/cn/home/index.html. The ICDRs data are available at the website of the official ROMSAF website, and the link is as below: https://rom-saf.eumetsat.int/product_archive.php. The (numeric) data underlying the results of this study are available from the corresponding author upon qualified request.*"

**Edits/Language**

Ln 21: Substitute "which are about" with "at about"

Ln 24: Consider shortening "required to improve the quality issue for improving" to just "for improving"

Ln 30: Substitute "As propagating" with "While propagating"

Ln 31: Not sure what "formulated" here means

Ln 33: By "of can" do you mean "of water wapour"?

Ln 35: "2022" should be "2002"

Ln 36: Remove "to be used"

Ln 39: "all weather" should be "all-weather"

Ln 42: "system" should be plural

Ln 44: Substitute "In the meanwhile" with "Meanwhile"

Ln 45: Substitute "detecting" with "detection"

Ln 47: "and also tropical cyclones" is missing a verb or noun (e.g., "detection", "identification")

Ln 70: Remove "few" from "In recent few years"

Ln 70: Remove comma after "(ROMEX)"

Ln 71: Substitute "of three months' period" with "over a three-month period"

Ln 74: Remove "improving"

Ln 76: Either "in stratospheric regions" or "in the stratospheric region"

Ln 77: "bending angles" should be "bending angles"

Ln 78: Substitute "There are current a lot of work" with "Several works"

Ln 87: "a couple of reports" is informal. Use instead "two"

Ln 87: Remove "in" from "in analysing"

Ln 89: "i.e.," should be "e.g.,"

Ln 90/91: Consider rewriting the sentence as: "Unlike the case of statistical optimization and 1dVar (Li et al., 2013, 2019), the retrieval of bending angles is not influenced by background models"

Ln 92: "are useful" should be "is useful"

Ln 92: Repetition: substitute the second instance of "understanding" with "assessing"

Ln 92: "subsequent retrieved" should be "subsequently retrieved"

Ln 96: "examination" should be "examining"

Ln 97: "subsequent retrieved" should be "subsequently retrieved"

Ln 98: Please list both parties of this joint work, not only CMA.

Ln 148: Remove ": QF1, QF2, and QF3 are used to detect outliers", since it's a repetition of the first part of the sentence.

Ln 161: "exceed threshold" should be "exceeds the threshold"

Ln 161-163: Please consider rephrasing as: "QF0 indicates a profile of good quality, i.e., where no quality issues have been detected. QF8 indicates that at least one of the seven quality flags has a non-zero value."

Ln 176: "observations" should probably be "latitudinal bands" or something similar.

Ln 182: "largest errors" should be "the largest errors"

Ln 213: "also include quality" should be "also includes the quality"

Ln 214: "set a strict criteria" should be "set as a strict criteria"

Ln 214: "solely use" should be "solely using"

Ln 255: "altitude" should be "latitude"

Ln 286: "are mainly comes" should be "mainly come" (or "are mainly coming")

Ln 287: "suggest a positive" should be "suggests positive"

Ln 288: "data are used" should be "data used"

Ln 313: "sources" should be "source"

Ln 362: "to 15%" should be "15%" (two instances on this line)

Ln 364/365: "biases are around" should be "biases around" (two instances)

Ln 376: "were given" should be "given"

**Overall responses on language edits:** Thanks really for these detail comments on language editing. They are really useful for non-native speakers. We have revised them in our updated manuscript.

We would like to thank Reviewer #1 again for correcting some erroneous information in our original manuscript and for helping us gain a deeper understanding of the quality of Fengyun data.

---

## Author Comment (AC2)

**Responses to Reviewer#2**

**Overall comments:** Review of "Quality aspects of Fengyun3 D/E radio occultation bending angle products" by Li et al. The manuscript presents an analysis of FY3 D/E bending angles through a comparison with ERA5 and MetOp observations. After applying quality control, the FY3 D/E results are found to be generally consistent with MetOp data below ~40 km. At high altitudes the FY3 D/E observations exhibit larger biases and noise. The results are clearly presented, illustrating the overall quality of the FY3 D/E data after application of quality control procedures. The overall topic is suitable for publication in AMT, and I recommend the paper for publication following revisions to address some aspects of the text that are unclear. Specific comments are provided below.

**Overall responses:** We thank Reviewer#2 for his/her overall comments. We have carefully addressed all the major and minor comments. Please see our responses below and also the revised manuscript. The line numbers refer to the track change version of the revised manuscript.

**Major Comments:**

**Major comment 1.** The comparisons are done against ERA5. The assimilation of RO observations in ERA5 could potentially impact the results. It is recommended that the authors clarify if the FY3 D/E data are assimilated in ERA5. The fact that the MetOp data are assimilated in ERA5 should also be stated in the text. Furthermore, the potential impact of the assimilation of these data in ERA5 on the results should be clarified in the text.

**Responses to major comment 1:** Thanks for this comment. In fact, we have checked that both MetOp data and FY3 data have been assimilated into ERA5. However, the impacts on our results can be minor for two reasons. First, assimilation of RO data are mostly below the middle stratosphere. However, this study mainly focus on the quality of bending angle at high altitudes (above the middle stratosphere). Secondly, due to relatively smaller number of MetOp and FY3 RO observations (compared to the overall quantity of RO profiles assimilated), their influences on the calculated ERA5 data are considered can be neglected.

Yes, in order to make this statement clearer, we have clarify this information in the data section, which can be seen from lines 309 to 312 in page 5 of the revised manuscript and also as below:

"*It should be noted that both MetOp data and FY3 data have been assimilated into ERA5. However, the impacts on our results can be minor for two reasons. First, assimilation of RO data are mostly below the middle stratosphere. However, this study*

*mainly focus on the quality of bending angle above middle stratosphere. Secondly, due to relatively smaller number of MetOp and FY3 RO observations, their influences on the calculated ERA5 data are considered can be neglected.*"

**Major comment 2.** In line 186, the authors state that the spikes that appear in the data "have been reduced in the latest version of Fengyun data." It would be beneficial to include some additional details on what was changed in the data processing to address this issue.

**Responses to major comment 2:** Thanks for this comment. Considering this comment and also a similar comment from Reviewer#1, we have corrected the original statements into the following (please also see lines 403 to 407 of page 7 in the revised manuscript):

*"A spike transition at about 25 km is observable in the error profiles of Fengyun satellites. This is primarily due to two reasons: 1) extrapolation of L2 signal at fixed 25 km; 2) fixed transition height at 25 km). Since 2025, this algorithm has been updated. First, the extrapolation height of L2 will be determined by the quality of L2 signal. Second, the fixed transition height was changed from 25 km to 15 km. By this update, the standard deviations of refractivity has been reduced by about 0.5% in this height range."*

**Major comment 3.** One of the conclusions based on the quality control is that MetOp bending angles have minimal quality issues, and that this is partly due to "strict internal quality control system integrated into the ROPP software" (lines 216-217 and 358). It would be useful to include additional details in terms of what is meant by this statement. Does this mean that the poor quality data were already removed in the processing?

**Responses to major comment 3:** Thanks for this comment. After reconsidering this statement, we think our original statement is not correct. In fact, FY3 series and MetOp data all use the ROPP software to process their RO data. Therefore, the quality control scheme should be similar. In the updated manuscript, we have deleted this statement. We have revised corresponding statements into:

"*This is attributed to its high-performance receiver of MetOp satellite.*"

**Minor Comments:**

**Minor comment 1:** Line 33: The statement "where the existence of can be neglected" is missing a word.

**Responses to minor comment 1:** Thanks for this comment. Yes, in the revised manuscript, we have revised this statement to "where the existence of water vapor can be neglected". Please see line 38 in the revised manuscript.

**Minor comment 2:** Line 63: "SPIRE" should be "Spire" (it is not an acronym).
**Responses to minor comment 2:** Thanks. In the revisied manuscript, we have corrected SPIRE to Spire.

**Minor comment 3:** Lines 78-79: The statement "There are current a lot of work…" should be revised as it is grammatically incorrect.
**Responses to minor comment 3:** Thanks. We have revised to "several works have analyzed…"

**Minor comment 4:** Line 94: What is "NOMEX"?
**Responses to minor comment 4:** Sorry for this typo. We have corrected to "ROMEX"

**Minor comment 5:** Lines 320 and 323: Should "K" be "%"?
**Responses to minor comment 5:** Thanks for pointing this out and also sorry for our typo. We have corrected to % in the updated manuscript. `

**Minor comment 6:** Line 348: "empirically analysis of" should be "empirical analysis of"
**Responses to minor comment 5:** Thanks, we have corrected to "empirical analysis of"

We thank Reviewer#2 again for his/her valuable comments to make our manuscript better.